# Antennal Transcriptome Analysis of Olfactory Genes and Characterization of Odorant Binding Proteins in *Odontothrips loti* (Thysanoptera: Thripidae)

**DOI:** 10.3390/ijms24065284

**Published:** 2023-03-09

**Authors:** Yanqi Liu, Yingning Luo, Lixiao Du, Liping Ban

**Affiliations:** 1College of Grassland Science and Technology, China Agricultural University, Beijing 100193, China; 2Institute of Plant Protection, Chinese Academy of Agricultural Sciences, Beijing 100091, China

**Keywords:** *Odontothrips loti* Haliday, antennae transcriptome, olfactory, odorant binding proteins

## Abstract

To identify odors in complex environments accurately, insects have evolved multiple olfactory proteins. In our study, various olfactory proteins of *Odontothrips loti* Haliday, an oligophagous pest that primarily affects *Medicago sativa* (alfalfa), were explored. Specifically, 47 putative olfactory candidate genes were identified in the antennae transcriptome of *O. loti*, including seven odorant-binding proteins (OBPs), nine chemosensory proteins (CSPs), seven sensory neuron membrane proteins (SNMPs), eight odorant receptors (ORs), and sixteen ionotropic receptors (IRs). PCR analysis further confirmed that 43 out of 47 genes existed in *O. loti* adults, and *O.lot*OBP1, *O.lot*OBP4, and *O.lot*OBP6 were specifically expressed in the antennae with a male-biased expression pattern. In addition, both the fluorescence competitive binding assay and molecular docking showed that *p*-Menth-8-en-2-one, a component of the volatiles of the host, had strong binding ability to the O.lotOBP6 protein. Behavioral experiments showed that this component has a significant attraction to both female and male adults, indicating that O.lotOBP6 plays a role in host location. Furthermore, molecular docking reveals potential active sites in O.lotOBP6 that interact with most of the tested volatiles. Our results provide insights into the mechanism of *O. loti* odor-evoked behavior and the development of a highly specific and sustainable approach for thrip management.

## 1. Introduction

*Odontothrips loti* Haliday (Thysanoptera: Thripidae) is an oligophagous insect that mainly attacks leguminous forages, especially *Medicago sativa* L. (alfalfa). *Odontothrips loti* damages the young tissues of alfalfa, including the leaves and flowers, with its piercing–sucking mouthparts during the adult and larval stages. The impact of thrips on alfalfa is significant, with damage rates of 70 to 100% reported in Hohhot, China [1]. *Odontothrips loti* is also known to transmit the alfalfa mosaic virus, one of the most serious pathogens in alfalfa planting, and seriously reducing the yield and quality of alfalfa hay [2]. Moreover, thrips usually lead a short life generation with high fecundity inside the flower bud, which greatly increases the challenges of prevention and control. The indiscriminate use of chemical insecticides has resulted in emerging resistance in several economically important thrips [3,4]. Therefore, ongoing research is focusing on alternative strategies, including biogenic pesticides [5], biological control [6], host plant resistance [7], semiochemicals interference [8], and gene editing [9], to achieve the more effective management of these pests.

Reducing population density by interfering with intra- and interspecific communication is considered to be an effective strategy to reduce pest impacts. In recent years, with the regularization of “omics” techniques, the specific behavior of insects has been elucidated at the genetic level, especially the olfactory neural mechanisms [10,11]. It is well known that olfactory perception through the peripheral nervous system captures and transports signal molecules to neurons to transform into nerve impulses, which are carried to the central nervous system to combine with other senses, to convert olfactory signals into behavioral responses [12]. At present, olfactory proteins involved in odorant molecular transduction in insects include odorant-binding proteins (OBPs), chemosensory proteins (CSPs), odorant receptors (ORs), ionotropic receptors (IRs), and sensory neuron membrane proteins (SNMPs) [13].

OBPs and CSPs are mainly distributed in the lymph of the olfactory sensilla, and they are the first components of the olfactory transduction cascade and are known as the solubilizers and carriers of hydrophobic pheromones and odorants [14]. Recent analyses of OBP9 in the wheat aphid *Sitobion avenae* F. indicate that they bind to multiple wheat volatiles, suggesting that this protein is involved in host localization functions [15]. A similar change could occur in the CSPs of *Oedaleus asiaticus* Bei-Bienko, given that three CSP proteins are involved in the detection of host volatiles and even the initiation of aggregation behavior [16]. In addition, they are highly expressed in tissues other than antennae and are involved in complicated physiological functions [17,18]. For example, OBPs in *Nilaparvata lugens* Stål were highly expressed in the salivary gland and involved in the host defense responses [19]. OBPs were also able to affect the reproductive capacity of *Aedes aegypti* Linnaeus [20] and improve the insecticide resistance of *Diaphorina citri* Kuwayama [21]. Similarly, CSPs have been reported to promote development in insects [22,23].

Odor stimuli released by OBPs or CSPs are recognized by two families of olfactory receptors, including ORs and IRs, located on the membranes of olfactory sensory neurons (OSNs) in peripheral olfactory organs [24]. They function as ligand-gated ion channels that converge chemical stimuli via OSNs axons in the neuropil region of the antennal lobe’s glomeruli, which is the basic functional unit of olfactory information processing [24,25]. While ORs are expressed primarily in basiconic and trichoid sensilla and detect volatile chemical signals, such as esters and alcohols [26], IRs are expressed primarily in coeloconic sensilla, which are multimodal receptive entities [25]. In addition to olfactory recognition, IRs are involved in taste sensing, temperature and humidity sensing, circadian rhythm synchronization, etc. [27,28]. More strikingly, these olfactory receptors establish a link with specific behavioral responses, providing a direct theoretical basis for pest control. For example, OR35 was responds specifically to 4-Vinylanisole (4VA), which has been identified as a pheromone in gregarious *Locusta migratoria* L. and regulates the aggregation behavior of locusts [29]; HassOR31 in *Helicoverpa assulta* Guenée was involved in determining precise egg-laying sites in host plants [30]; OR47b, OR88a, OR67d, and OR19a act in the mating, aggression, or oviposition of *Drosophila melanogaster* Meigen [31,32,33,34]. The function of IRs depends on the expression of widely expressed coreceptors, such as IR8a or IR25a in *D. melanogaster* [35]. IR8a and IR64a have been shown to form a functional complex in vivo, which mediates the recognition of acids [36], while IR8a in *H. armigera* Hübner mediated attraction behavior to acetic acid [37]. SNMPs are localized on the membrane of OSNs in peripheral olfactory organs and belong to the CD36 superfamily [38]. SNMPs worked synergistically with ORs or ORs/Orco complexes and were necessary for the recognition of pheromone and general molecular odor [11,39].

Recently, these olfactory proteins have allowed the development of genetic tools to visualize and manipulate specific olfactory pathways to determine how odors are transmitted to evoke behavior. The assessment of the ligand binding properties of olfactory proteins, particularly OBPs or CSPs, is essential for understanding their physiological functions. Fluorescence competition assay provides an initial insight into protein binding properties by calculating the dissociation constants of fluorescent probes and protein complexes in competition with ligands and is widely used in studies of insect olfactory proteins [40,41]. With the development of molecular dynamics studies, such as homology modelling and molecular docking, it is possible to accurately predict the binding properties of olfactory proteins, making the detection of behaviorally active pest compounds more efficient [40,42].

Our knowledge of olfactory protein families in Thysanoptera is, however, still scarce. To date, olfactory genes information in thrips are still confined to *Frankliniella occidentalis* Pergande, *Thrips palmi* Karny, and *F. intonsa* Trybom [43,44] but is essential for understanding the olfactory recognition of thrips and their induced behaviors.

In this study, transcriptomics analysis were used to explore candidate olfactory genes in *O. loti* antennae, followed by the functional detection of O.lotOBP6, a typical OBP, based on its binding properties, molecular docking, and olfactory behavior. Our results may provide insights into the mechanism of their odor-evoked behavior and the development of a highly specific and sustainable approach for thrip management.

## 2. Results

### 2.1. Transcriptome Sequencing and Functional Annotation

To identify olfactory genes, the transcriptome sequencing of female antennae of *O. loti* was undertaken. Approximately 91.79 million clean reads were generated. A total of 38,276 unigenes were obtained by Trinity with a mean length of 1113 bp, of which 10,404 (27.18%) were more than 1 kb in length, and N50 of 2294 bp (Appendix A). The length distribution of unigenes is listed at Appendix A.

The functional annotation of unigenes was performed, including comparison with the NR, Swiss-prot, COG, KOG, KEGG, GO, and Pfam protein databases. The results indicate that 16,788 (43.86%) unigenes were annotated in the above databases (e-value ≤ 1 × 10^−5^). Among them, 16,697 (43.62%) unigenes matched the NR database (Table 1) and 9397 (24.55%) unigenes are present in three major databases (NR, Swiss-prot, KOG) (Appendix A). Homology analysis with other species in the NR database showed that *F. occidentalis*, a widely distributed pest thrips, shared the highest identity (21.38%) with *O. loti* (Figure 1A). In addition, GO annotation was conducted to categorize the function of transcripts, and a total of 6276 unigenes were annotated in three functional groups, namely cellular component, molecular function, and biological process (Figure 1B). In the KEGG annotation, 3478 unigenes were successfully annotated into six types of pathways, among which 714 genes were annotated into “signal transduction” pathways, accounting for 20.53% of the total (Figure 1C).

### 2.2. Screening and Identification of Olfactory Candidate Genes

A total of 47 putative olfactory candidate genes were identified in the antennae transcriptome of *O. loti*, including seven odorant-binding proteins, nine chemosensory proteins, eight odorant receptors, seven sensory neuron membrane proteins, and sixteen ionotropic receptors. Most genes contain complete ORF except for *O.lot*IR6, 9, 13, 15, 16, and *O.lot*SNMP1, 2, 7. The ORF sequence of the gene has been uploaded to GenBank, and the accession numbers of each sequence are shown in Table 2, Appendix A. In addition, according to the structural characteristics of the protein, the number of N-terminal signal peptides and transmembrane domains was predicted (Table 2, Appendix A).

There are generally six conservative cysteine residues in the insect OBPs sequence, and the pairwise combination forms three cross-linked disulfide bonds, which is also an important basis for identifying the OBP family. All *O.lot*OBPs identified had six conserved cysteine residues, except for *O.lot*OBP7, which had four conserved cysteine residues, possibly belonging to the Minus-C OBPs subfamily (Figure 2A). To show the homologous relationships of all putative *O. loti* OBPs with other insect gene sets, a phylogenetic tree was constructed using the protein sequences of 111 OBPs from twelve species, comprising two thrips, nine species of Hemiptera, and *D. melanogaster* as an outgroup (Figure 2B). In terms of species affinities, *O.lot*OBPs are more closely related to Thysanoptera. The phylogenetic analysis demonstrated that *O.lot*OBP7 belongs to the Minus-C subfamily of OBPs (Appendix A), which is consistent with the multiple sequence alignment (Figure 2A). It is notable that *O.lot*OBP2 belongs to the Atypical OBPs subfamily, in that, with the exception of six conserved cysteine residues, there is one positively conserved cysteine residue on each side of the sequence, consistent with the sequence characteristics of the Atypical OBPs subfamily (Appendix A). Nine *O.lot*CSPs in total were identified with four relatively conserved cysteine residues, which is in accord with CSP sequences in other insect species (Appendix A). Phylogenetic analysis showed that most of the *O.lot*CSPs were scattered in three clades, which contained almost all the known Thysanoptera CSPs (Appendix A).

Based on olfactory candidate proteins from Thysanoptera and Hemiptera, phylogenetic analysis of three membrane-bound proteins (O.lotORs, O.lotIRs, and O.lotSNMPs) was studied (Appendix A). The results show that *O.lot*OR5 was clustered into the odorant co-receptors (ORco) subfamily, suggesting it could function as a complex with other ORs in the *O. loti (*Appendix A). In the phylogenetic analysis of *O.lot*IRs, IR21a, IR25a, IR40a, IR93a, and IR75 orthologues were identified in *O. loti*, all of which are highly related to the IRs identified in other Thysanoptera (Appendix A). Our results showed that *O.lot*IR1, 3, 4, 10, 11, 13, and 16 were clustered into the IR75a subfamily, suggesting that they may be involved in the recognition of acids and amines [45]; *O.lot*IR8, 12, and 15 were clustered into the IR25a subfamily, suggesting that they could function as co-receptors of IRs [35,46]; *O.lot*IR5, 7, and 14 were clustered into the IR93a subfamily; *O.lot*IR6 and 9 were clustered into the IR21a subfamily; and *O.lot*IR2 was clustered into the IR40a subfamily, suggesting that it may participate in the perception of temperature and humidity [47,48]. In addition, most of the *O.lot*SNMPs were scattered in three clades, which contained almost all the known Thysanoptera SNMPs (Appendix A).

### 2.3. Specific Expression Level of Olfactory Candidate Genes

The expression of 47 olfactory candidate genes in the antennae and residual body parts of female and male adults, respectively, were verified by RT-PCR, of which 43 were verified except *O.lot*OBP3, *O.lot*OR3, *O.lot*IR3, and *O.lot*IR6 (Figure 2C). The results showed that all of the 43 genes were differentially expressed in antennae compared to the residual body parts. It is worth noting that some genes were specifically expressed in the antennae, including *O.lot*OBP1, 4, and 6; *O.lot*OR1, 2, and 5; *O.lot*IR1, 8, 9, and 10; and *O.lot*SNMP4 and 5 (Figure 2C), illustrating their potential olfactory recognition functions. Among these, notable differences in expression were also observed between the sexes; for example, female antennae were significantly more expressed than males in *O.lot*IR10 and *O.lot*SNMP4.

To further investigate typical OBPs specifically expressed in antennae, the expression of three OBPs (*O.lot*OBP1, 4, 6) in different tissues and developmental stages was monitored by RT-qPCR (Figure 2D,E). The expression of *O.lot*OBP1, *O.lot*OBP4, and *O.lot*OBP6 in male antennae was 1.73, 3.97, and 11.69 times higher compared to female antennae, respectively, which was consistent with the trend found using RT-PCR verification. The temporal expression profiles of *O.lot*OBP1, 4, and 6 showed that male adults had the highest levels, followed by the 2nd instar nymph, which may be related to the active feeding at this stage. The expression level in pupa was greatly reduced (Figure 2E) and was usually inactive in the 3rd and 4th instar nymphs.

### 2.4. Fluorescent Binding Assay of O.lotOBP6

In order to further verify the function of O.lotOBPs, we selected O.lotOBP6 and expressed it in a prokaryotic expression system and the purified recombinant O.lotOBP6 protein was obtained, which was consistent with the expected purified molecular weight of 25.21 kDa (Appendix A). Using N-phenyl-1-naphthylamine (1-NPN) as a fluorescent probe in the binding assay, we determined the binding affinities of 16 volatile substances to recombinant O.lotOBP6 protein (Table 3). The binding curve of 1-NPN to recombinant O.lotOBP6 is shown in Figure 3A, and the dissociation constant of recombinant O.lotOBP6 protein to 1-NPN was 1.05 umol/L, as calculated using the Scatchard equation.

The competitive fluorescence binding curves showed that 11 ligands, namely, nonanal, m-xylene, tridecane, eucalyptol, *p*-anisaldehyde, salicylaldehyde, 2-ethyl-1-hexanol, *p*-Menth-8-en-2-one, (Z)-3-hexenyl butyrate, linalool, and benzaldehyde, showed high binding affinities (K_i_ < 10 µM) with the O.lotOBP6/1-NPN complex (Figure 3B,C and Table 3). Notably, *p*-Menth-8-en-2-one showed strong competitive ability, suggesting that O.lotOBP6 protein plays an important role in host location, as *p*-Menth-8-en-2-one was one of component identified from leguminous forages *M. sativa* leaves.

### 2.5. Homology Modeling and Molecular Docking of O.lotOBP6

The 3D model of O.lotOBP6 was constructed based on A.aegOBP1 (PDB: 3k1e.1.A) of *A. aegypti*. The Ramachandran plot showed 90.0% of residues in the favored regions, and 100.0% of all residues were in allowed regions (Appendix A). The Verify_3D results showed that 98.2% of O.lotOBP6 residues scored above 0.2 (Appendix A), indicating that the protein model was successfully constructed. The O.lotOBP6 contains six α helices (Figure 4A), which are located between residues Ala19 and Thr36 (α1), Tyr40 and Asp48 (α2), Glu54 and Cys67 (α3), Leu78 and Lys84 (α4), Asn91 and Thr94 (α5), and Leu105 and Lys119 (α6). It is predicted that there are three potential hydrophobic binding pockets in the structure of O.lotOBP6, all of which are located around the hydrophobic amino acid residues (Figure 4B). The sizes of these binding pockets are 284 Å^3^, 103 Å^3^, and 24 Å^3^ in volumes, and their volume depths are 1517 Å, 323 Å, and 101 Å, respectively.

In order to further verify the results of fluorescence competitive binding test, the molecular docking simulation of 11 successful competitive ligands was carried out (Figure 4C–E and Appendix A, Table 3). The results showed that there were nine ligands, m-xylene, eucalyptol, *p*-anisaldehyde, salicylaldehyde, 2-ethyl-1-hexanol, *p*-Menth-8-en-2-one, (Z)-3-hexenyl butyrate, linalool, and benzaldehyde, bound to pocket 2 with binding energies of −4.34, −5.32, −4.76, −4.53, −3.64, −5.65, −4.64, −4.07, and −4.67, respectively. Nonanal binds to pocket 3 with the binding energy is −3.86. Tridecane binds to pocket 1, and the binding energy is −4.04. Ligands can form hydrogen bonds, hydrophobic interactions, and even perform Π-stacking with multiple amino acids. In conclusion, the high affinity of *p*-Menth-8-en-2-one with ligand was consistent with the results of fluorescence competitive binding assay, and pocket 2 was speculated to be the binding site of O.lotOBP6 to *p*-Menth-8-en-2-one.

### 2.6. Choice Tests with Volatiles

Y-tube olfactometer assays were performed with the 11 effective chemical ligands based on fluorescence binding assay. The results showed that both female and male *O. loti* were significantly attracted by *p*-Menth-8-en-2-one, a substance derived from fresh *M. sativa* leaves (Liu Y-Q, unpublished data [52]), compared to the control (Figure 5). This result demonstrated that O.lotOBP6 is involved in the recognition of *p*-Menth-8-en-2-one and plays a role in the host plant localization. In addition, male *O. loti* were significantly attracted by (Z)-3-hexenyl butyrate, a substance detected in alfalfa flowers [51].

## 3. Discussion

Multiple olfactory protein families are involved in the insect peripheral olfactory system, which are important for their survival and reproduction [53]. The olfactory proteins and their potential value in pest control have been well documented in Lepidoptera [54,55], Hymenoptera [56,57], Diptera [58], Orthoptera [59], Coleoptera [11,60,61], and Hemiptera [62,63,64] but rarely in Thysanoptera [43,44]. However, our knowledge of the olfactory protein families in Thysanoptera does require further research. By screening the genome data of *F. occidentalis* and *T. palmi*, currently the only two species with genome data available in Thysanoptera, we compared the candidate genes for olfactory sense in the three species (Table 4). The number of ORs and IRs in *F. occidentalis* is significantly more than that of the other two species of thrips, which may be related to its higher environmental adaptability, especially regarding host diversity. *Frankliniella occidentalis* is known to be a polyphagous invasive pest and is recorded as feeding on more than 250 crop species from over 60 families [65]. Conversely, *T. palmi* is mainly associated with plants from Solanaceae and Cucurbitaceae families [66] and *O. loti* mainly to the Leguminosae [67]. Furthermore, more than 18 plant volatiles were reported to be attractive to *F. occidentalis* [50]. This suggests that a highly evolved chemosensory proteins plays an important role in the adaptability of thrips population [47].

We analyzed the transcriptome data of female antennae of *O. loti*, and verified 43 olfactory genes, including six OBPs, nine CSPs, seven ORs, seven SNMPs, and fourteen IRs. Among them, *O. lot*OBP6 was highly expressed in both female and male antennae, and the function was further validated. The deduced amino acid sequences suggested that *O.lot*OBP6 consists of a typical framework of OBPs (six-conserved cysteines), which was confirmed by phylogenetic analysis (Figure 2). In terms of species affinities, the protein is more closely related to the Thysanoptera species. The first Thysanoptera insect OBPs were identified from *F. occidentalis*, predominantly expressed in antennae [43]. Here, we provide the first insight into the function of the OBPs family in the Thysanoptera, based on ligand-binding properties and behavioral evidence. Fluorescence binding assays are a common method to evaluate the ligand-binding properties of OBPs [68], but there is the possibility of false positives, which do not provide structural information about the protein binding to the ligand [69]. With the development of molecular dynamics, homologous modeling and molecular docking make up for this shortcoming. Based on protein 3D structure and molecular docking, a large number of potential semiochemicals were identified, for example, β-ionone to BtabOBP3 in *Bemisia tabaci* Gennadius [41], octyl methoxycinnamate, dibutylphthalate, myristic acid and palmitic acid to TrufOBP4 in *Tirathaba rufivena* Walker [40] and E10-16: Ald to CpinPBP2 in *Conogethes pinicolalis* Inoue and Yamanaka [70]. In our study, *p*-Menth-8-en-2-one was screened due to its strong binding ability to *O. lot*OBP6.

Research focused on understanding the interaction between plant secondary metabolites and insects is an important aspect to developing effective pest management strategies. There is growing evidence that OBPs specifically expressed in antennae are involved in long-distance chemical communication [71,72]; for example, the antenna-specific OBP, *Bdorsobp2* in *Bactrocera dorsalis* Hendel, is implicated in the recognition of methyl eugenol, a potential species-specific attractant, based on RNAi, electrophysiological, and behavioral evidence [73], while *BodoOBP1* and *BodoOBP2*, which are specifically expressed in male antennae of *Bradysia odoriphaga* Yang et Zhang, have been demonstrated to participate both in host localization and courtship behavior [74]. A similar situation was observed in our study of O. lotOBP6, where both fluorescent binding assay and choice assays demonstrated the strong interaction of this protein with *p*-Menth-8-en-2-one, which was mainly found in *Mentba L*. and extensively used as spice [75].

It is worth mentioning that (Z)-3-hexenyl butyrate were not detected in alfalfa leaves (Liu Y-Q, unpublished data) but were present in alfalfa flowers [51]. (Z)-3-hexenyl butyrate was also detected in the volatiles of *Chrysanthemum lavandulifolium* and *Flemingia macrophylla* leaves [76,77]. This substance has been reported to elicit a strong electrophysiological and behavioral response by *Dasychira baibarana* Matsμmura [78] and induces the selection of egg-laying sites by *H. assulta* [30]. This volatile may be one of key factors of the interaction between alfalfa and *O. loti*.

Various chemical compounds have been found to attract thrips and have been developed into attractants [79,80]. For example, compared to the control, (2E,6E)-farnesyl acetate, a component secreted by *Megalurothrips usitatus* Bagrall males, was significantly more attractive to adults in field trials and at a dose of 60 μg, this effect lasted at least 6 days [81]. Using *p*-Menth-8-en-2-one and (Z)-3-hexenyl butyrate in lures might provide new opportunities for thrips management, but more effort is being devoted to dosing, stabilizers, and combinations with other semiochemicals to achieve a stable and sustained effect in the field.

In conclusion, we investigated the function of O.lotOBP6, which is specifically expressed in the antennae and has a strong affinity for a host volatile, *p*-Menth-8-en-2-one. This study contributes to the systematic study of olfactory genes of Thysanoptera. Future functional studies of these antennae OBPs will provide new insights into the molecular basis and evolution of chemical reception. To date, the antennal sensilla morphology data of *O. loti* have been obtained [82] and work is in progress to determine the function of *O. lot*OBP6 through immunolocalization and single sensillum recording studies, and the potential pheromones of *O. loti* are being verified in parallel. We would speculate that these proteins are not only involved in interspecific chemosensing but also act as carriers of pheromones, as reported for several other species [14,74].

## 4. Materials and Methods

### 4.1. Insect Rearing and Antenna Collection

The adult *O. loti* used in this study were collected in July 2020 from the leaves of *M. sativa* growing in a stand located at the China Agricultural University, Beijing, China (40°1′42″ N, 116°16′43″ E). They were reared in the laboratory with *M. sativa* leaves at 26 ± 1 °C, 60 ± 5% RH under a photoperiod of 16 h light/8 h dark. For transcriptome analysis, around 1000 pairs female adult antennae were dissected, transferred to 1.5 mL microcentrifuge tubes, frozen in liquid nitrogen, and stored at −80 °C until use.

### 4.2. RNA Extraction and Llumina Sequencing

Approximately 1000 pairs of adult female antennae were used for RNA extraction. The total antennal RNA was extracted using TRIzol Reagent (cat. no.15596026, Invitrogen, Waltham, MA, USA), following the manufacturer’s standard protocol. Extracted RNA was quantified using a NanoDrop ND1000 spectrophotometer (Thermo Scientifc, Waltham, MA, USA). Quality-checked RNA samples were sequenced by Illumina HiSeq 2500 System (NovoGene, Shanghai, China). The total antennae RNA samples were sent to NOVOGENE for RNA quality testing, library construction, and sequencing at a depth of 10G.

### 4.3. Sequence Assembly and Functional Annotation

Clean reads were obtained from the raw reads after filtering out both low quality reads and the sequence reads containing adapters and poly-A/T tails. The resulting clean reads were assembled to produce unigenes with Trinity using the default parameters. All unigenes obtained from *O. loti* were annotated against the non-redundant protein sequence database (Nr), Swiss-prot protein sequence database (Swiss-prot), clusters of orthologous groups for eukaryotic complete genomes (KOG), and the database of clusters of orthologous genes (COG) using BLAST with a significant cut-off e-value of <1 × 10^−5^. The HMMSCAN program was used for annotation with the Pfam database. The blast results with Nr were further imported into the Blast2GO pipeline for Gene Ontology (GO) annotation. KAAS (https://www.genome.jp/tools/kaas/, accessed on 20 April 2021) was used for annotation with the Kyoto Encyclopedia of Genes and Genomes (KEGG) database.

### 4.4. Screening and Identification of Olfactory Candidate Genes

Based on our reference antennae transcriptome assembly, OBPs, CSPs, SNMPs, ORs, and IRs genes were retrieved from the *O. loti* unigenes. All candidate genes were manually examined using the BLASTx program with the Nr database. Using the open reading frame of olfactory candidate genes predicted by ORF Finder (https://www.ncbi.nlm.nih.gov/orffinder, accessed on 13 May 2021), the protein sequence was compared with the Pfam database (http://pfam.sanger.ac.uk, accessed on 13 May 2021), and the accuracy of the open reading frame was further determined by the matched family information. SignalP 5.0 (https://www.cbs.dtu.dk/services/signalP, accessed on 16 May 2021) was used to predict the N-terminal signal peptide. The prediction of protein transmembrane domain by TMHMM2.0 (http://www.cbs.dtu.dk/services/TMHMM, accessed on 20 May 2021) and multiple sequence alignment by Clustal Omega (https://www.ebi.ac.uk/Tools/msa/clustalo, accessed on 20 May 2021). Phylogenetic trees were constructed using the maximum-likelihood method with 1000 bootstrap replicates in IQ-TREE (v1.6.12) [83] and visualized in the Interactive Tree of Life (iTOL) [84]. WebLogo 3 (https://weblogo.threeplusone.com/, accessed on 6 January 2022) was used to generate the sequence logo. 

### 4.5. PCR Analysis

Active male and female adults were dissected under a dissecting microscope with a scalpel blade to precisely separate the antennae from the rest of the insect (hereafter referred to as residual bodies). Fifty antennae and residual bodies of male and female adults, respectively, as well as 10 nymphs each of the 1st instar and 2nd instar, 10 pupa (including a mixed sample of pre-pupae of the third instar and pseudopupa of the fourth instar), 10 female or male adults were collected separately, transferred to 1.5 mL microcentrifuge tubes, frozen in liquid nitrogen, and stored at −80 °C until use.

Total antennae RNA was extracted using TRIzol Reagent (cat. no.15596026, Invitrogen, Waltham, MA, USA) following the manufacturer’s standard protocol. The RNA was reverse transcribed to cDNA using a PrimeScript RT Reagent Kit with gDNA Eraser (cat. no. RR047A, TaKaRa, Beijing, China) according to the manufacturer’s instructions, after which the cDNA was stored at −20 °C. For RT-PCR, 2× Taq PCR Master Mix (cat. no. KT201-01, Tiangen, Beijing, China) was used for amplification reaction according to the manufacturer’s instruction with specific primers (Appendix A) designed by Premier 6.0. Amplifications were performed in a 25 μL reaction mixture containing 0.5 μL of cDNA, 1 μL of F/R primers (10 μM), and 12.5 μL of 2× Taq PCR Master Mix. The PCR conditions were 3 min at 94 °C, followed by 15 cycles of 10 s at 94 °C, 10 s at 65 °C (Each cycle was reduced by 1 °C), and 10 s at 72 °C, followed by 20 cycles of 10 s at 94 °C, 10 s at 50 °C, and 10 s at 72 °C, and then a final extension of 5 min at 72 °C. A total of 5 μL of the PCR product was analyzed by 1% agarose gel electrophoresis. For RT-qPCR, TB Green Premix Ex Taq II (Tli RNaseH Plus) (cat. no. RR820, TaKaRa, Beijing, China) was used for amplification reaction on an Analytikjena qTOWER3 Real-time PCR thermocycler (Analytikjena, Jena, Germany). Amplifications were performed in a 25 μL reaction mixture containing 10 ng of cDNA, 0.5 μL of F/R primers (10 μM), and 12.5 μL of TB Green Premix Ex Taq II. The PCR conditions were 5 min at 95 °C, followed by 40 cycles of 10 s at 95 °C, 30 s at 60 °C, and 15 s at 72 °C. The relative quantification of the real-time PCR data was performed using the comparative 2^−ΔΔCt^ method. A one-way analysis of variance was conducted to examine the intergroup differences in the mRNA levels of the target genes. Graphs were created using GraphPad Prism 7.0 software.

### 4.6. Preparation of Recombinant O.lotOBP6

Specific primers were designed based on the open reading frame (ORF) sequence of O.lotOBP6. The forward primer was 5-GGATCCGCCGCGCCCGCT-3 (the BamHI restriction site is underlined), and the reverse primer was 5-AAGCTTGGACGAGCCGGGGAGC-3 (the HindIII restriction site is underlined). Specific primers used for RT-PCR, methods and procedures are described in Section 4.5. PCR products were verified using 1% agarose gel. Gel pure was performed using the HiPure Gel Pure DNA Mini Kit (cat. no. D2111, Magen, Guangzhou, China) following the manufacturer’s standard protocol. The purified PCR product of O.lotOBP6 was cloned into a pTOPO001 Simple Vector (cat. no. TC601, Genesand, Beijing, China), promptly followed by the transformation of the resulting product into competent *E. coli* (DH5α) (cat. no. SCC12, Genesand, Beijing, China), according to the manufacturer’s instructions. The bacterial solution was transferred to solid Luria–Bertani (LB) medium containing 100 μg/mL ampicillin for inverted culture at 37 °C overnight. A 2× T5 Super PCR Mix (Colony) (cat. no. TSE005, Tsingke, Beijing, China) was used to identify the genes of single bacteria. Amplifications were performed in a 25 μL reaction mixture containing 1 μL of single bacteria, 1 μL of F/R primers (10 μM), and 12.5 μL of 2× T5 Super PCR Mix. The PCR validation of positive colonies was performed on LB medium containing 100 μg/mL ampicillin at 37 °C overnight with shaking at 250 rpm, and further sequencing was performed. The extraction of plasmid was carried out using high purity plasmid DNA small extraction kit (cat. no. T-PM0201, Tsingke, Beijing, China), according to the manufacturer’s instruction.

The plasmid was digested with the BamHI and HindIII enzyme at 37 °C, the excised target gene was purified, and ligated into the expression vector pET-30a (+) (MiaoLing Plasmid Platform, Wuhan, China). The recombinant plasmid was transformed into competent *E. coli* BL21 (DE3) (cat. no. SEC14, Genesand, Beijing, China) and the bacterial solution was transferred to solid LB medium containing 50 µg/mL kanamycin for inverted culture at 37 °C overnight. Single colonies were grown overnight in LB medium at 37 °C with shaking at 200 rpm. The culture was extended culture until the OD600 reached 0.6~0.8, while β-D-1-thiogalactopyranoside (IPTG) was added at a final concentration of 0.1 mM. After 3 h at 37 °C, the cells were harvested by centrifugation and resuspended in 1× PBS (pH 7.2~7.4) lysed by sonication. Protein was purified using His-tag Protein Purification Kit (Denaturant-resistant) (cat. no. P2229S, Beyotime, Shanghai, China). Purified protein was concentrated using Amicon^®^ Ultra-15 10KD centrifugal filters (Millipore, Darmstadt, Germany) and analyzed by SDS-PAGE. The CDS sequence and protein sequence of O.lotOBP6 were uploaded to GenBank under accession numbers OM732434.1 and WBU77199.1, respectively.

### 4.7. Fluorescence Binding Assays

Fluorescence was measured on a F-7000 fluorescence spectrophotometer (Hitachi, Tokyo, Japan) at room temperature with a 1 cm light path quartz cuvette and 10 nm slits for both excitation and emission. The fluorescent probe *N*-phenyl-1-naphthylamine (1-NPN) was excited at 337 nm, and emission was recorded between 370 and 520 nm. The pure protein was dissolved with 1× PBS (pH 7.2~7.4) at a final concentration of 2 μM. The ligands and 1-NPN were dissolved in methanol at the concentration of 1 mM.

To measure the affinity of 1-NPN to O.lotOBP6, a 2 μM solution of protein in 2 mL 1× PBS was titrated with 1-NPN to final concentrations of 0.25~8 μM. The affinity of ligands was measured by the titration of the O.lotOBP6/1-NPN complex at the concentration of 2 μM by adding ligand to final concentrations of 0.25~10 μM. The dissociation constants of the competitors were calculated by the equation K_i_ = IC_50_/(1 + [1-NPN]/K_1-NPN_), where IC_50_ is the concentration of ligands halving the initial fluorescence value of 1-NPN, [1-NPN] is the free concentration of 1-NPN, and K_1-NPN_ is the dissociation constant of the complex O.lotOBP6/1-NPN. Experiments were performed in triplicates. GraphPad Prism 7.0 was used to perform nonlinear regression analysis on the standard curve and the dissociation curve.

### 4.8. Homology Modeling and Molecular Docking

The 3D structure prediction of O.lotOBP6 was completed on the platform SWISS-MODEL (https://swissmodel.expasy.org, accessed on 24 January 2022). The template with the highest similarity was generally selected as the reference template, and the *A.aeg*OBP1 (PDB:3k1e.1.A) of *A. aegypti* was chosen as the reference template in this case, with a sequence similarity of 29.09% to O.lotOBP6. The 3D model quality assessment was performed using SAVES v6.0 (https://saves.mbi.ucla.edu, accessed on 24 January 2022). Docking ligand molecules were downloaded from the chemical compound databases ZINC (https://zinc.docking.org/, accessed on 5 February 2022) and TCMSP (https://old.tcmsp-e.com/molecule.php?qn=610, accessed on 5 February 2022). The POCASA 1.1 program (https://g6altair.sci.hokudai.ac.jp/g6/service/pocasa/, accessed on 5 February 2022) was used to make a pocket prediction, the grid size was set to 1 Å, and the atomic type was set to all. AutoDock 4.2 was used to find the potential binding mode between O.lotOBP6 and ligands. The grid box was set to cover the entire protein model, and the number of *x*, *y*, and *z* dimensions were 94, 104, and 102, respectively. The grid spacing is 0.375 Å. The *x*, *y*, and *z* center grid box was 7.975, 41.953, and 20.579. OpenBabel was used for format conversion. Visual structure analysis was carried out by PYMOL Viewer and PLIP (https://plip-tool.biotec.tu-dresden.de/plip-web/plip/index, accessed on 20 February 2022).

### 4.9. Choice Tests with Volatiles

In the behavioral assays, the female and male adults were reared together, while 3–5 days old female and male adults were collected separately by reproductive morphology. Eleven ligands with high binding affinities to O.lotOBP6/1-NPN complex based on fluorescence binding assays were used in the test. The responses of female and male *O. loti* to 11 volatiles (Table 3) were measured derived from *M. sativa* (Liu Y-Q, unpublished data) or semiochemicals previously used in Y-tube olfactometer bioassays [49,50,51]. The glass Y-tube consisted of a stem and two arms (10 cm long, 1 cm in diameter) separated from each other at an angle of 75°. Air drawn into the olfactometer was first purified by active charcoal, moistened by distilled water in a gas-washing bottle, and split into two streams, with a flow rate of 50 mL min^−1^. Two glass flasks (100 mL) provided test and control odor sources. Connections between the components of the olfactometer apparatus were made with Teflon tubes. Olfactometer experiments were carried out in a dark room at 26 ± 1 °C. Light was provided by a LED light (28 W), which was placed at the Y end of the olfactometer. The volatile sample of 10 ng/μL was dissolved in paraffin oil as the test odor source, with the control sample being paraffin oil. A single *O. loti* was introduced to the base tube of the Y-tube and observed for 5 min. When the test thrip entered a half-length of either arm of the Y-tube and stayed for 1 min, the choice was recorded. “No choice” was recorded if a test thrip did not make enter an arm during the bioassay period. After five test thrips were tested, the odor sources were swapped to avoid directional effects in the apparatus. To avoid the effects of travelling residual odors of test thrips, ten test thrips tests were followed by a new Y-tube, which was cleaned with 75% ethanol and oven dried at 60 °C for 30 min. Sixty biological replicates were carried out for each female or male treatment. Results were analyzed by Chi-square test using IBM SPSS 22.

## Figures and Tables

**Figure 1 ijms-24-05284-f001:**
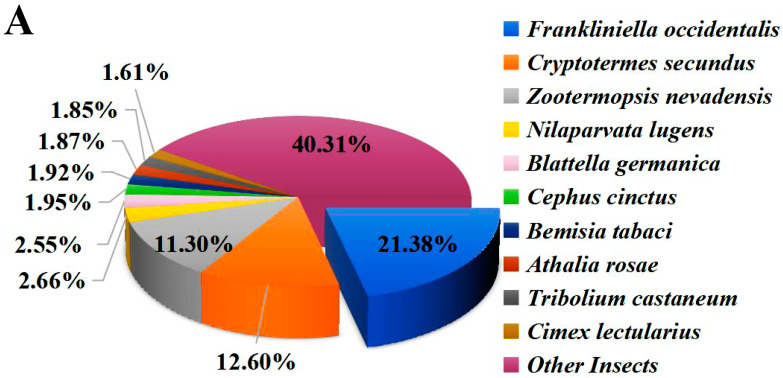
Functional annotations of *O. loti* unigenes. (**A**) The homology analysis of *O. loti* unigenes compared to other insect species. (**B**) Gene ontology (GO) classification of the *O. loti* unigenes. (**C**) The Kyoto encyclopedia of genes and genomes (KEGG) pathway enrichment analysis of *O. loti* unigenes.

**Figure 2 ijms-24-05284-f002:**
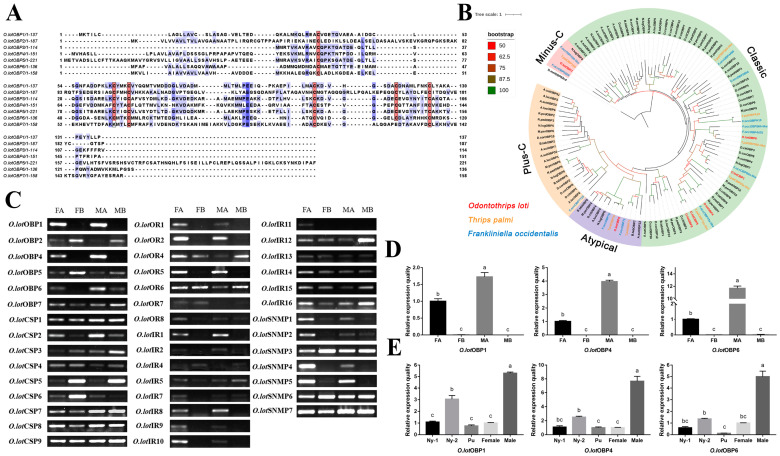
(**A**) Sequence alignment of *O.lot*OBPs. Red boxes show conserved cysteines, and the color of the purple box shows the strength of homology, the darker the color, the higher the homology. (**B**) The phylogenetic relationship between the OBP family and related species of *O. loti*. (**C**) Expression profiles of olfactory candidate genes in different tissues of *O. loti*. (**D**,**E**) Relative expression levels of *O.lot*OBP1, *O.lot*OBP4, and *O.lot*OBP6 in different tissues (**D**) and developmental stages (**E**). Significance level is indicated by lowercase letters, having the same letter means the difference is not significant, otherwise the difference is significant. Standard errors are indicated by error bars. *O.lot*: *Odontothrips loti*; *F.occ*: *Frankliniella occidentalis*; *T.pal*: *Thrips palmi*; *R.ped*: *Riptortus pedestris*; *A.sut*: *Adelphocoris suturalis*; *B.tab*: *Bemisia tabaci*; *D.mel*: *Drosophila melanogaster*; *M.per*: *Myzus persicae*; *R.pad*: *Rhopalosiphum padi*; *A.luc*: *Apolygus lucorum*; *N.lug*: *Nilaparvata lugens*; *D.cit*: *Diaphorina citri*; *D.cor*: *Drosicha corpulenta*; FA: female antennae; FB: female residual body tissue excluding the female antennae; MA: male antennae; MB: male residual body tissue excluding the male antennae; Ny-1: 1st instar nymph; Ny-2: 2nd instar nymph. Pu: pupae; Female: female adult; Male: male adult.

**Figure 3 ijms-24-05284-f003:**
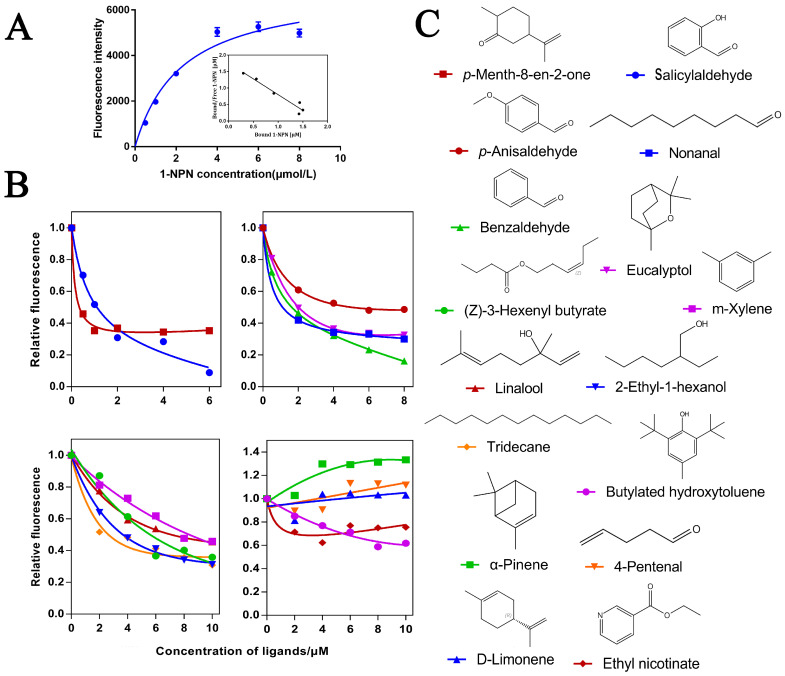
Affinity of recombinant O.lotOBP6 to candidate chemical ligands. (**A**) Binding curves and Scatchard plots of 1-NPN to O.lotOBP6. (**B**) Competition binding curves of O.lotOBP6 and ligands. (**C**) The chemical structures of the chemical ligands.

**Figure 4 ijms-24-05284-f004:**
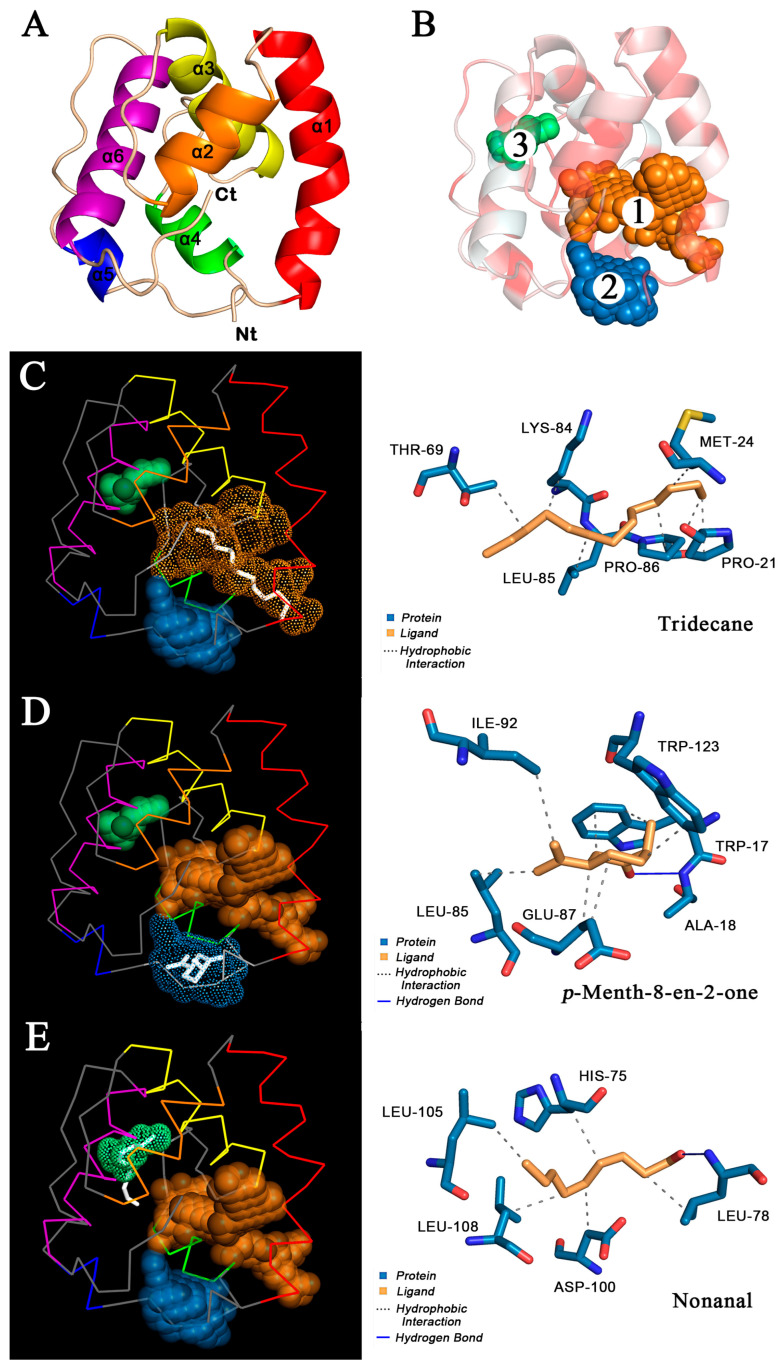
Homology modelling and molecular docking of O.lotOBP6. (**A**) The 3D structure of O.lotOBP6 was drawn in SWISS-MODEL based on the reference template *A.aeg*OBP1 (PDB:3k1e.1.A) of *A. aegypti*. The O.lotOBP6 contains six α helices, differently colored. (**B**) Hydrophobic properties and predictive binding pockets of O.lotOBP6. The darker the red, the stronger the hydrophobicity. The three binding pockets are represented by spheres of different colors and numbered according to volume from largest to smallest. (**C**–**E**) Molecular docking of O.lotOBP6 with tridecane, *p*-menth-8-en-2-one, and nonanal. Ligand molecules are shown in white (**left**)/yellow (**right**), and amino acid residues are shown in blue (**right**). The six α helices of O.lotOBP6 are represented by lines of different colors, consistent with those in Figure 4A.

**Figure 5 ijms-24-05284-f005:**
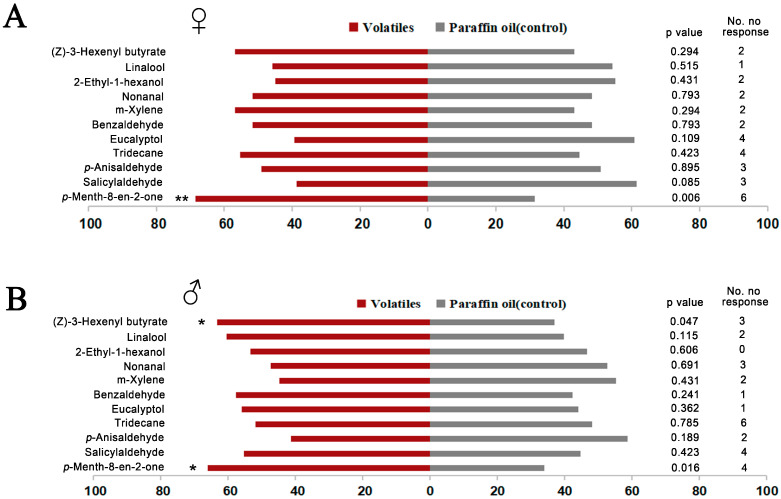
Responses of adult *O. loti* to 11 volatiles in a Y-tube olfactometer. In the assays, newly emerged 60 female or male adults, of 3–5 days of age, were used for testing, and 11 ligands with high binding affinities to O.lotOBP6 based on fluorescence binding assays were used in the test. Results were analyzed by the chi-square test using IBM SPSS 22. (**A**) Responses of female adult *O. loti*. (**B**) Responses of male adult *O. loti*. Asterisks indicate significant differences within a choice test (* *p* < 0.05; ** *p* < 0.01).

**Table 1 ijms-24-05284-t001:** Functional annotation of the unigenes associated with the antennae of female *O. loti* and recovered from seven databases.

Annotated Database	Annotated Number	300 ≤ Length < 1000	Length ≥ 1000	Percentage of Annotated Unigenes/%
NR_Annotation	16,697	7885	8812	43.62
Swiss-prot_Annotation	11,042	4517	6525	28.85
KOG_Annotation	11,463	3941	7521	29.95
GO_Annotation	6276	3578	2698	16.40
KEGG_Annotation	6118	1009	5109	15.98
COG_Annotation	5271	2837	2434	13.77
Pfam_Annotation	295	115	16	0.20
Total unigenes	16,788	7951	8837	43.86

**Table 2 ijms-24-05284-t002:** Summary of odorant-binding proteins (OBPs) and chemosensory proteins (CSPs) sequences identified in female antennae of *O. loti*.

Gene Name	Gene ID	GenBank Acc. Number	ORF (bp)	Complete ORF	Signal Peptide	Homology Search with Known Proteins			
						Species	Acc. Number	e-Value	Identity
*O.lot*OBP1	c37901.graph_c0	OM732430	414	Yes	1–19	*F. occidentalis*	XP_026290294.1	1 × 10^−68^	76.92%
*O.lot*OBP2	c45609.graph_c1	OM732431	564	Yes	1–19	*F. occidentalis*	XP_026283043.1	7 × 10^−93^	74.10%
*O.lot*OBP3	c64409.graph_c0	--	345	Yes	0	*D. melanogaster*	NP_524242.2	6 × 10^−8^	24.11%
*O.lot*OBP4	c27233.graph_c0	OM732432	456	Yes	1–23	*F. occidentalis*	XP_026290809.1	1 × 10^−58^	73.23%
*O.lot*OBP5	c25468.graph_c0	OM732433	666	Yes	0	*T. palmi*	XP_034238437.1	2 × 10^−28^	45.30%
*O.lot*OBP6	c38184.graph_c0	OM732434	411	Yes	1–18	*T. palmi*	XP_034232373.1	1 × 10^−40^	52.17%
*O.lot*OBP7	c39971.graph_c0	OM732435	477	Yes	1–19	*T. palmi*	XP_034238274.1	6 × 10^−30^	43.92%
*O.lot*CSP1	c47703.graph_c0	OM732436	405	Yes	1–19	*F. occidentalis*	AEP27186.1	3 × 10^−76^	83.58%
*O.lot*CSP2	c15858.graph_c0	OM732437	390	Yes	1–19	*Dendroctonus armandi*	AXF53965.1	8 × 10^−37^	40.17%
*O.lot*CSP3	c29971.graph_c0	OM732438	387	Yes	1–19	*Dioryctria abietella*	QJX59186.1	2 × 10^−42^	57.43%
*O.lot*CSP4	c35792.graph_c0	OM732439	465	Yes	1–21	*F. occidentalis*	AKF17719.1	8 × 10^−59^	76.24%
*O.lot*CSP5	c36473.graph_c0	OM732440	483	Yes	0	*Heortia vitessoides*	AZB49393.1	1 × 10^−37^	43.94%
*O.lot*CSP6	c40571.graph_c0	OM732441	456	Yes	1–25	*Oedaleus infernalis*	AYN71370.1	2 × 10^−50^	81.40%
*O.lot*CSP7	c41025.graph_c0	OM732442	414	Yes	1–18	*Riptortus pedestris*	AWW17227.1	3 × 10^−47^	55.45%
*O.lot*CSP8	c47707.graph_c0	OM732443	411	Yes	1–22	*Vespa crabro*	AAV68929.1	1 × 10^−24^	33.64%
*O.lot*CSP9	c47720.graph_c0	OM732444	372	Yes	1–20	*Papilio xuthus*	BAF91715.1	5 × 10^−24^	32.14%

**Table 3 ijms-24-05284-t003:** Binding ability of recombinant O.lotOBP6 to ligands identified in host plants.

Chemicals Classification	Ligands	IC_50_ (μmol/L)	K_i_ (μmol/L)	Binding Energies	Binding Pocket
Aldehydes	Nonanal	1.40 ± 0.09	0.58 ± 0.04	−3.86	3
	4-Pentenal	>50	--	--	--
	Salicylaldehyde	1.10 ± 0.01	0.45 ± 0.00	−4.53	2
	*p*-Anisaldehyde *	4.59 ± 0.64	1.89 ± 0.26	−4.76	2
	Benzaldehyde *	1.62 ± 0.12	0.67 ± 0.05	−4.67	2
Alcohols	2-Ethyl-1-hexanol	3.80 ± 0.31	1.57 ± 0.13	−3.64	2
	Linalool	6.92 ± 0.24	2.86 ± 0.10	−4.07	2
Terpenoids	D-Limonene	>50	--	--	--
	α-Pinene	>50	--	--	--
Ketones	*p*-Menth-8-en-2-one	0.34 ± 0.03	0.14 ± 0.01	−5.65	2
Esters	(Z)-3-Hexenyl butyrate *	5.58 ± 0.38	2.30 ± 0.16	−4.64	2
Hydrocarbons	Tridecane	2.60 ± 0.44	1.07 ± 0.18	−4.04	1
Aromatic Hydrocarbons	m-Xylene	8.41 ± 0.31	3.47 ± 0.13	−4.34	2
Phenols	Butylated hydroxytoluene	>50	--	--	--
Ethers	Eucalyptol *	2.06 ± 0.05	0.85 ± 0.02	−5.32	2
Nitrogenous compounds	Ethyl nicotinate *	>50	--	--	--

IC_50_: The concentration of ligands halving the initial fluorescence value of 1-NPN in fluorescence binding assays. K_i_: dissociation constant (K_i_ = IC_50_/(1 + [1-NPN]/K_1-NPN_)). Binding energies and binding pocket: summary of molecular docking of O.lotOBP6 to ligands. * Chemicals showing behavior reaction to thrips [49,50,51].

**Table 4 ijms-24-05284-t004:** Quantitative statistics of olfactory candidate genes found in *O. loti* compared to *F. occidentalis* and *T. palmi*.

Gene	*Odontothrips loti*	*Frankliniella occidentalis*	*Thrips palmi*
OBPs	6	12	8
CSPs	9	3	n.a.
ORs	7	84	15
IRs	14	>167	6
SNMPs	7	4	4

n.a. = No relevant annotated genes in the genomic data. OBPs: odorant-binding proteins; CSPs: chemosensory proteins; ORs: odorant receptors; IRs: ionotropic receptors; SNMPs: sensory neuron membrane proteins.

## Data Availability

The antennal transcriptome sequencing data has been deposited in the Genome Sequence Archive (GSA) database under Accession Number CRA008641.

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
