# Peer review of "Antennal Transcriptome Analysis of Olfactory Genes and Characterization of Odorant Binding Proteins in Odontothrips loti (Thysanoptera: Thripidae)"

_ijms, 2023, doi:10.3390/ijms24065284_

Round 1
Reviewer 1 Report
Reviewer Comments #: -The manuscript describes “Antennal transcriptome analysis of olfactory genes and characterization of odorant binding proteins in Odontothrips loti (Thysanoptera: Thripidae)”. The author tries to analyze olfactory genes and characterize OBP in O. loti. Techniques used in the manuscript are up to date and the experiments are well performed. However, I found certain things which need clarification to support and strengthen their conclusion.
Abstract needs modification. Please make it simple.
Please check the typos in the manuscript.
Lines 72-93 need to be reorganized and possibly place in the latest references that suit the manuscript. The last paragraph needs to be polished.
Section 4.2: Please provide details about the antenna how many etc.
What do you mean by residual bodies in your manuscript?
Section 4.9: “emerged female and male adults 3-5 days” are they rared separately or together? Please provide such information which is missing.
In general, a few details in MM are missing in the text please incorporate especially in section 4.6
The third paragraph of the discussion needs attention please revise it and the last paragraph of the discussion should be improved from the reader's point of view.
A few parts of the discussion are weak please update the references for a better explanation.
Please add the latest references to the introduction since most of the references are old please update them.
A few parts of the discussion are weak please update the references for a better explanation.
Details on figure legends are missing. Please provide complete information.
General question:
Why do not you use the negative controls for your experiments on your gels?
Do you ever try these primers on different tissues like legs, fat body, brain, etc?
Please properly input the details overall to get the full attention of the reader for example the connection between the thrips and O.loti
General comments:
Please correct English throughout your article. I recommend going for a native speaker of a company.
There are many typos in the present form of the manuscript please correct them before submission.
A graphical abstract can provide a better understanding.
Conclusion: The manuscript needs some corrections, I still ask the authors to rearrange and extend information on certain parts of the manuscript, especially the introduction and the discussion. I do think that the manuscript contains important issues, information, interesting approaches, and techniques, which can lead to a proper understanding of the role of olfactory genes and OBP in pest management strategies. So I consider this manuscript suitable for publication after the suggested major revision in IJMS.
Author Response
Response to Reviewer 1 Comments
Point 1: The manuscript describes “Antennal transcriptome analysis of olfactory genes and characterization of odorant binding proteins in Odontothrips loti (Thysanoptera: Thripidae)”. The author tries to analyze olfactory genes and characterize OBP in O. loti. Techniques used in the manuscript are up to date and the experiments are well performed. However, I found certain things which need clarification to support and strengthen their conclusion.
Response 1: Thanks for the comments, and we really appreciate it. In response to your valuable comments, we have made the suggested changes to improve readability. These are highlighted in red text.
Point 2: Abstract needs modification. Please make it simple.
Response 2: Thanks for the suggestion! We have revised the Abstract (see lines 10-24).
Point 3: Please check the typos in the manuscript.
Response 3: Thank you for your valuable advice. We have proofread the text and addressed any typos.
Point 4: Lines 72-93 need to be reorganized and possibly place in the latest references that suit the manuscript. The last paragraph needs to be polished.
Response 4: Thanks for the suggestion! The third paragraph of the introduction has been rearranged to quote as much of the latest literature as possible. And the last paragraph was refined to hopefully address your concerns.
Point 5: Section 4.2: Please provide details about the antenna how many etc.
Response 5: Thanks for the suggestion! For transcriptome analysis, about 1000 pairs female adult antennae were collected, and this information was added in line 377. Other details have been refined accordingly.
Point 6: What do you mean by residual bodies in your manuscript?
Response 6: Residual bodies is the thrips tissue minus the antennae. This section has been added, see section 4.5 and legend of figure 2.
Point 7: Section 4.9: “emerged female and male adults 3-5 days” are they reared separately or together? Please provide such information which is missing.
Response 7: Thank you for your valuable advice. The female and male adults that emerged were reared together and this section has been added, see line 508.
Point 8: In general, a few details in MM are missing in the text please incorporate especially in section 4.6
Response 8: Thanks for the suggestion! We have added details to the MM section, especially Section 4.6.
Point 9: The third paragraph of the discussion needs attention please revise it and the last paragraph of the discussion should be improved from the reader's point of view.
Response 9: Thanks for the comments. The third paragraph has been revised and the language polished, and the last paragraph has also been revised. Please see lines 332-367.
Point 10: A few parts of the discussion are weak please update the references for a better explanation.
Response 10: Thank you for your valuable advice. We have revised the discussion section as a whole, adding molecular docking and fluorescence binding tests, citing the latest references, and polishing the language.
Point 11: Please add the latest references to the introduction since most of the references are old, please update them.
Response 11: Thanks for the suggestion! We updated the references in the introduction, citing as many as possible the latest reference.
Point 12: Details on figure legends are missing. Please provide complete information.
Response 12: Thank you for your valuable advice. We have improved figure legends in the text for better reading and understanding, especially Figures 4 and 5.
Point 13: Why do not you use the negative controls for your experiments on your gels?
Response 13: Thanks for the suggestion! We apologize for not designing a negative control, the agarose gel results are a preliminary validation and qRT-PCR validation has been performed using negative controls, and thus can rule out false positives.
Point 14: Do you ever try these primers on different tissues like legs, fat body, brain, etc?
Response 14: Thanks! We did not try PCR amplication with other tissues as the template, apart from the antennae. Being a small insect, thrips bodies are approximately 2-5 mm long, which make it challenging to dissect and precisely collect different tissues.
Point 15: Please properly input the details overall to get the full attention of the reader for example the connection between the thrips and O.loti.
Response 15: Thanks for the suggestion! We have corrected the writing details to clearly distinguish between thrips and O. loti, for example line 278, line 372 etc.
Point 16: Please correct English throughout your article. I recommend going for a native speaker of a company.
Response 16: Thanks for the suggestion! We sought the help of native English speakers to polish the manuscript. And we had thoroughly edited the manuscript, especially for English writing, including language and grammar/syntax et al.
Point 17: There are many typos in the present form of the manuscript please correct them before submission.
Response 17: Thank you for your valuable advice. We have made formatting corrections throughout the text to correct these spelling errors.
Point 18: A graphical abstract can provide a better understanding.
Response 18: Thank you for your valuable advice. We have submitted a graphic abstract to the journal editor as shown below.

Reviewer 2 Report
General comment:
This work has been skilfully done and the manuscript has been thoroughly prepared. All in all, a fine work. There are only some few corrections to be done.
Specific comments:
p. 1, line 36: please change to "... piercing-sucking mouthparts ..." (accoerding to Gullan & Cranston, The Insects - An outline of Entomology)
p. 2, line 44: replace "natural enemy application" by "biocontrol" or "biological control"
p. 2, line 67: "Stål" instead of "Stal"
p. 3, line 119: "..., a widely distributed pest thips, ..."
p. 4, figure 1: in the present form, the text and the figures are hard to read. I recommend to enlarge the figures and put figure 1A and B on one page and Fig. 1C and the legend on the next.
p. 5, figure 2: similar recommendation as for fig. 1, maybe the enlarged figure fits better on a cross page.
p. 6, line 177: correct spacings and comma between terms in parentheses
p. 7, line 188: "... that they may participate in perception ..."
p. 9, figure 4 and Supplementary materials, figure S12: enlarge full figure, and put text in 4B describing the colors in larger letters
p. 10, legend to fig. 4, line 257: "... helices, differently colored."
p. 10, legend to fig. 5: please give full information of statistical procedure used
p. 15, lines 548-549: insert full stop each after "XXI" and "VII"
p. 15-18, References: please check whether the titles of cited literature are set in lowercase (except proper names), e.g. ref. 9, change to "Neuroethology of olfactory-guided behavior ....."
Supplementary materials:
Check figure S4 to S9, they are difficult to read, larger letters could help
Figure S11 should be enlarged
Author Response
Response to Reviewer 2 Comments
Point 1: This work has been skilfully done and the manuscript has been thoroughly prepared. All in all, a fine work.
Response 1: Thank you for your appreciation of our work.
Point 2: line 36: please change to "... piercing-sucking mouthparts ..." (according to Gullan & Cranston, The Insects - An outline of Entomology)
Response 2: Thank you for your valuable advice. This has been corrected, see line 30.
Point 3: line 44: replace "natural enemy application" by "biocontrol" or "biological control"
Response 3: Thanks for the suggestion! This has been corrected, see line 39.
Point 4: line 67: "Stål" instead of "Stal"
Response 4: Thanks for the suggestion! This has been corrected, see line 61.
Point 5: line 119: "..., a widely distributed pest thips, ..."
Response 5: Thanks for the suggestion! It has been modified to “..., a widely distributed pest thrips, ...”, see line 120-121.
Point 6: figure 1: in the present form, the text and the figures are hard to read. I recommend to enlarge the figures and put figure 1A and B on one page and Fig. 1C and the legend on the next.
Point 7: figure 2: similar recommendation as for fig. 1, maybe the enlarged figure fits better on a cross page.
Response 6-7: Thank you for your valuable advice. We have enlarged Figure 1 and Figure 2, increased the size of part of the font, and split Figure 1 into two pages.
Point 8: line 177: correct spacings and comma between terms in parentheses
Response 8: Thanks for the suggestion! This has been corrected, see line 181. We have made formatting corrections throughout the text to avoid such errors.
Point 9: line 188: "... that they may participate in perception ..."
Response 9: Thanks for the suggestion! This has been corrected, see line 191-192.
Point 10: figure 4 and Supplementary materials, figure S12: enlarge full figure, and put text in 4B describing the colors in larger letters
Response 10: Thank you for your valuable advice. We have enlarged the figure 4 and S12 for better reading. Figure 4B has been amended accordingly.
Point 11: legend to fig. 4, line 257: "... helices, differently coloured."
Response 11: Thanks for the suggestion! This has been corrected, see line 266.
Point 12: legend to fig. 5: please give full information of statistical procedure used
Response 12: Thanks for the suggestion! The explanation of this part is supplemented on lines 281-284.
Point 13: lines 548-549: insert full stop each after "XXI" and "VII"
Response 13: Thanks for the suggestion! This has been corrected, see line 587-588.
Point 14: References: please check whether the titles of cited literature are set in lowercase (except proper names), e.g. ref. 9, change to "Neuroethology of olfactory-guided behavior ....."
Response 14: Thank you for your valuable advice. All the titles of cited literature have been set to lower case (except proper names).
Point 15: Check figure S4 to S9, they are difficult to read, larger letters could help
Point 16: Figure S11 should be enlarged
Response 15-16: Thanks for the suggestion! We have enlarged and bolded the text to improve readability and hopefully the clarity of the image.

Reviewer 3 Report
In this work, Liu and co-workers highlight the role of OBPs in olfactory recognition, mating behavior and host locations, which is very interesting. My review focused on docking studies, of which I leave the followig comments:
-In Introduction section the authors did not end it with an insight into the computational results or contribution, as they did in the Abstract.
- I would rather see Table S4 in the main text, perhaps combined with Table 3. Note that the binding energy generally follows the Ki trend for the collection of molecules. This is very good for docking validation in predicting affinity. The discussion will gain with comments in this regard.
-In Figure 4, woul be helpful to see the label (3-letter code) of the amino acids around the odorant molecule, as for the identification of a pattern of consistent interactions.
In Methods:
4.8 -> "OBP6 was completed..." using what sequence? what UniProt code?
line 468: what AutoDock? version, Vina, AutoDock 4...?
line 468: "pocket prediction" - > So, oriented-docking was performed, choosing coordinates of each pocket??? What was the grid box dimension, grid spacing...?
Why blind docking was not considered?
I believe that by adding these informations, the work will be more easily reproducible and complete.
Author Response
Response to Reviewer 3 Comments
Point 1: In this work, Liu and co-workers highlight the role of OBPs in olfactory recognition, mating behaviours and host locations, which is very interesting.
Response 1: Thanks for the comments, and we really appreciate it.
Point 2: In Introduction section the authors did not end it with an insight into the computational results or contribution, as they did in the Abstract.
Response 2: Thanks for the comments! We modified the introduction part and added the description of fluorescence competitive binding experiment and molecular docking to make the content more complete. These changes can be viewed on lines 90-98.
Point 3: I would rather see Table S4 in the main text, perhaps combined with Table 3. Note that the binding energy generally follows the Ki trend for the collection of molecules. This is very good for docking validation in predicting affinity. The discussion will gain with comments in this regard.
Response 3: Thank you for your valuable advice. Table S4 has been merged with Table 3, see Table 3. This part of the discussion has also been added. Please see lines 322-331.
Point 4: In Figure 4, would be helpful to see the label (3-letter code) of the amino acids around the odorant molecule, as for the identification of a pattern of consistent interactions.
Response 4: Thanks for the suggestion! The corresponding labels have been added to Figure 4 and the same changes have been made to Figure S12.
Point 5: In Methods:
4.8 -> "OBP6 was completed..." using what sequence? what UniProt code?
Response 5: The CDS sequence and protein sequence of OBP6 were uploaded to Genbank under accession numbers OM732434.1 and WBU77199.1, respectively. The explanation of this part is added on lines 473-474.
Point 6: line 468: what AutoDock? version, Vina, AutoDock 4...?
Response 6: Thanks for the suggestion! It has been modified to AutoDock 4.2, Please see line 501.
Point 7: line 468: "pocket prediction"- > So, oriented-docking was performed, choosing coordinates of each pocket??? What was the grid box dimension, grid spacing...?
Point 8: Why blind docking was not considered?
Response 7-8: Thanks for the suggestion! Pocket prediction is completed using POCASA online tool, which is based on the rolling probe sphere protein pocket recognition algorithm for blind prediction [1]. We set the parameters as follows: the grid size is 1 Å, the atom type is all, and retain the default values for other parameters. The explanation of this part is supplemented on lines 500-501.
Molecular docking was completed using Autodock 4.2. The technique combines conformational search with a rapid grid-based method of energy evaluation, blind docking was carried out to find the early optimal binding position, rather than oriented-docking [2]. The grid box is set to cover the entire protein model, and the number of x, y and z dimensions are 94, 104, 102. The grid spacing is 0.375 Å. The x, y, z center grid box is 7.975, 41.953, 20.579. The explanation of this part is supplemented on lines 502-504.
Molecular docking and pocket prediction are done independently and the two models are then presented in the same visualization interface to verify the accuracy of the results. The results of molecular docking could help further select the correct pocket. In this paper, most of the ligand molecules appear in pocket No. 2, suggesting that pocket No. 2 is the correct pocket of O.lotOBP6.
Reference:
- Yu, J.; Zhou, Y.; Tanaka, I.; Yao, M. Roll: a new algorithm for the detection of protein pockets and cavities with a rolling probe sphere. Bioinformatics 2010, 26, 46-52, doi:10.1093/bioinformatics/btp599.
- Morris, G.M.; Goodsell, D.S.; Halliday, R.S.; Huey, R.; Hart, W.E.; Belew, R.K.; Olson, A.J. Automated docking using a Lamarckian genetic algorithm and an empirical binding free energy function. J Comput Chem 1998, 19, 1639-1662, doi:10.1002/(SICI)1096-987X(19981115)19:14<1639::AID-JCC10>3.0.CO;2-B.

Round 2
Reviewer 1 Report
No comments